# Is Sea Level Rise a Known Threat? A Discussion Based on an Online Survey

**Stefano Solarino [1],\*** , **Elena Eva [1]**, **Marco Anzidei [2]** , **Gemma Musacchio [3]** **and Maddalena De Lucia [4]**

1 Istituto Nazionale di Geofisica e Vulcanologia, 16145 Genova, Italy
2 Istituto Nazionale di Geofisica e Vulcanologia, 00143 Roma, Italy
3 Istituto Nazionale di Geofisica e Vulcanologia, 20133 Milano, Italy
4 Istituto Nazionale di Geofisica e Vulcanologia, 80124 Napoli, Italy
\* Correspondence: stefano.solarino@ingv.it

**Abstract:** Since the last century, global warming has been triggering sea level rise at an unprecedented rate. In the worst case climate scenario, sea level could rise by up to 1.1 m above the current level, causing coastal inundation and cascading effects, thus affecting about one billion people around the world. Though widespread and threatening, the phenomenon is not well known to citizens as it is often overshadowed by other effects of global warming. Here, we show the results of an online survey carried out in 2020–2021 to understand the level of citizens' knowledge on sea level rise including causes, effects, exacerbation in response to land subsidence and best practice towards mitigation and adaptation. The most important result of the survey is that citizens believe that it is up to governments to take action to cope with the effects of rising sea levels or mitigate the rise itself. This occurs despite the survey showing that they actually know what individuals can do and that a failure to act poses a threat to society. Gaps and preconceptions need to be eradicated by strengthening the collaboration between scientists and schools to improve knowledge, empowering our society.

**Keywords:** sea level rise; survey; best practice; adaptation; mitigation; coastal inundation; Mediterranean coasts





## 1. Introduction

Sea level rise (SLR) is a major consequence of global warming that is causing the melting of global ice and the thermal expansion of the oceans.

This phenomenon is worldwide affecting low elevation coastal zones, islands and littoral urban areas (large megacities as well as small villages), where about 1 billion people live. Coastal sites are undergoing coastal retreat and erosion, with relevant socioeconomic effects on human activities. Although the effects of rising sea levels can drastically change coastal areas in the long run and affect human activities, as has already happened in past centuries [1], the accelerated rise in sea level in the coming decades is still considered a minor risk by most coastal populations [2].

In the Earth's geological past, sea level changes due to astronomical phenomena-driven climate change have occurred several times [3]. However, the increase in global temperatures and global mean sea level (GMSL) which started about 150 years ago is undoubtedly due to human activities, according to the latest reports of the Intergovernmental Group on Climate Change, IPCC "www.ipcc.ch (accessed on 20 September 2023)".

The GMSL is rising at unprecedented rates with expected progressive inundation of the coastal zone [4] and with compelling consequences that are only a small part of the public agenda or debate [5–7].

Scientific data from ground and space instrumental observations show that the mean SLR of the oceans increased from 1.4 mm/year in the 20th century to about 3.7 (3.2–4.2) mm/year over the period 2006–2018, and will likely reach 5.2–12.1 mm/year in the period

2080–2100 for the lowest and highest $CO_2$ emission scenarios, respectively. This will lead to an expected upper limit of GSLR (global sea level rise) of about 1.1 m by the end of this century [8], which exceeds previous estimates published in the IPCC AR5 report (Figure 1) (updated after our survey by the publication of the AR6 report).

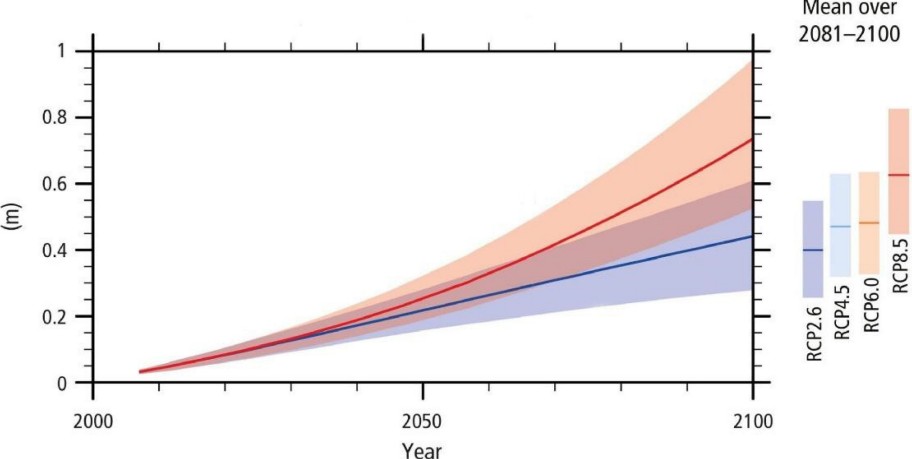

**Figure 1.** Global mean sea level rise from 2006 to 2100 relative to 1986–2005 for lowest (RCP2.6 in blue) and highest (RCP8.5 in red) projected emissions with related uncertainties (shaded colors). Modified from Climate Change 2014 Synthesis Report Fifth Assessment Report (AR5) Intergovernmental Panel on Climate Change at "https://ar5-syr.ipcc.ch/topic_summary.php (accessed on 5 September 2023)".

However, this limit may be higher due to the still unknown instabilities of the Greenland and Antarctic ice sheets [8]. According to [9], the ongoing phenomenon in the Mediterranean basin has several key components that can alter SLR estimates at a regional level.

Such an unprecedented rate of global mean sea level (GMSL) growth has compelling consequences that are not sufficiently addressed by the public agenda or debate. SLR is still often considered a minor risk by the coastal population, although the scientific data obtained from multiple disciplines ranging from climate to Earth sciences and biology agree in showing the global scale of the phenomenon. Earthquakes or volcanic eruptions may be very destructive, but they affect only limited areas of the Earth's surface, even during the strongest events. Conversely, SLR is a global phenomenon that can affect in time the coasts of each continent and island of the world, as well as populations who have been living close to the coastline since historical times [1]. Decision-makers and individuals are not sufficiently aware of the associated risks to take appropriate mitigation and adaptation policies [2,10].

In order to understand the reasons why a global emergency is coupled with ineffective actions, it is urgent to know to what extent the general public is informed about SLR, its effects and impacts, and even more importantly, to what extent there are misconceptions.

Here, we show results from an online survey carried out in the frame of the SAVEMEDCOASTS-2 project "www.savemedcoasts2.eu (accessed on 20 September 2023)" to evaluate the impacts of SLR along targeted sites of the Mediterranean coasts up to the year 2100, providing SLR projections and potential scenarios of coastal marine inundation, also in storm surge conditions, including the contribution of land subsidence along the coastal zone.

The aim of the survey was to support prevention and preparation actions in the Mediterranean coastal communities, through the knowledge of the phenomenon, necessary to deal with the effects and the socio-economic impact of sea level rise. In particular, our survey focused on five Mediterranean zones: the Venice lagoon and the coastal plain of Metaponto (Italy), the mouths of the Basento and Bradano rivers (Italy), the delta of the Ebro river (Spain), the coastal plain of Chalastra (Greece), Cyprus and the coast of Alexandria in the Nile delta (Egypt) and the Rhone delta (France).

To this end, people were asked to fill in a specific questionnaire published for a specific time window online at "www.savemedcoasts2.eu (accessed on 20 September 2023)". The questionnaire was designed and developed to understand the level of awareness of the investigated coastal communities

## 2. SLR Survey

Preparing coastal communities to address and mitigate the impacts of rising sea levels and to undertake adaptation strategies and prevention actions is an important and difficult task to achieve.

The goal is not only to show and understand future SLR scenarios in specific localities, but also to disseminate scientific results to the public and foster best practice. Whatever the risk-related theories, frameworks and models one may choose for the implementation of risk communication, the understanding of what the public knows and/or think about a certain risk is mandatory, and yet not a common practice [11]. It allows us, for instance, to implement effective risk communication that encourages action by the general public to limit risks and choose preparedness.

The public—or non-experts in general—may not be well enough informed or simply not care about a natural phenomenon. Generally speaking, it is thus of paramount importance to evaluate the knowledge of people about the causes and the effects of long-lasting phenomena, such as SLR, to set up the level of information and dissemination so as to improve prevention actions and adaptation planning. Although there are many publications about the SLR perception of the public around the world [1,10,12–22], there are still only a few studies of the Mediterranean area [2,9,23,24]. The phenomenon has only recently been taken into account as a consequence of the increased awareness of climate change.

We thus designed a survey in four languages that was published online and open to the general public. The English version of the questionnaire is shown in Figure S1 of the Supplementary Material. The questionnaire has been published in two forms: one for those who already know about SLR and one for those who do not. There are slight differences between the two questionnaires: in the first case the respondents are also asked about their source of information about the issue, while in the second case, since the respondents do not know about the phenomenon, the questions aim to elicit an opinion based on common sense and not on knowledge. The comparison between the answers of the two groups of respondents can help to estimate how much the knowledge of the SLR helps to foster best practice.

The survey is organized in three blocks: the first aims to know if the reader is at least aware of the rise in sea levels and, in that case, where they obtain the information; the second block asks about the causes and the consequences of SLR, who has responsibility for mitigating the effects, how to adapt our cities to the threat and what can be done to reduce SLR; the last block collects respondents' personal information regarding age, education, employment, vicinity to the coast of their home. The final field is left free for the respondents to comment on the survey or the phenomenon.

The answers in the questionnaire were designed after a careful revision of the content of the principal textbooks used in the schools of the countries involved in the survey and an analysis of the citizens' needs.

### The Respondents

We spread the request to compile the survey by word of mouth, soliciting teachers and writing a few posts on social networks. We also profited from dissemination by the press agencies of the institutions involved in the project. Our target has been to inquire about perceptions and knowledge of SLR to a wide population of the "generic public", without any restrictions of age, education or employment category.

Given that we did not impose any selection to the recruitment of the respondents, we can consider ours as a totally random sample. Random sampling is often used in science to conduct randomized control tests or for blinded experiments. Each individual of a

population set has the same probability of being included in the sample. This creates, in most cases, a balanced subset that carries the greatest potential for representing the larger group as a whole. Conversely to other sampling methods or in reference to a specific population (for example, all adults aged 25–60 and in higher education), we do not/can not compute the appropriate sample size like in [25,26]. All results and relative speculations must be then considered at a qualitative level.

The total number of respondents was 1454 from 23 countries, with particular feedback from the Mediterranean countries. However, the collected answers go far beyond, and give us the chance to obtain information also from countries that are not yet experiencing the phenomenon. In the next sections, we will first describe the sample and then we will discuss the answers and the findings.

One advantage of a random sampling approach is that we may guess that most respondents were really willing to contribute in a frank manner since they freely agreed to join in. However, this does not avoid vandalism. We then made a wide search for fake completions (by will or by chance) based on the coherency between age, job position or education level of the respondents. We assumed that a scammer does not pay attention to the way he/she fills out the fields of the questionnaire, giving them incoherent answers. If the respondent declares to be 17 and owns a PhD or is a teacher, we can flag this completed survey as suspect and remove it. A more demanding search was conducted for cloned completions by the same respondents. For example, in case of students from the same school, living in the same town and having the same age, some of the answers in the third block in the questionnaires were identical and, thus, suspect. Only the cross-checking of all answers permits us to discriminate whether they are multiple completions from the same respondent. It may of course happen, by chance, that two students input exactly the same answers: in these cases, both questionnaires were deleted. The net number of completions after the checking for not reliable entries is 1417, that is the 97% of total respondents.

In 7 out of 23 countries, more than 10 answers were collected. As expected and foreseen, most of the respondents compiled the questionnaires from the Mediterranean countries; the greatest number of completions was from Italy (992). Table 1 shows the number of respondents from each one of the 23 countries. In most cases, the number of answers does not allow us to check the dependency between level of knowledge and country of residence.

One piece of information missing from our analysis is the fraction of respondents who came across the questionnaire by chance, for example, by reading a press release about the experiment or a post on social media. We estimate that about 30% of the answers were compiled by people not directly solicited by friends, colleagues or teachers. As already stated, and in the frame of a random sampling, in an experiment like ours the optimal sample would be made only of people who were not directly invited to participate. However, we believe that the way respondents have been involved is not biased, since it does not imply that they are more informed. It may have some geographical influence on the number of respondents living in coastal areas if the solicitors themselves live there. However, it must be remarked that such a number may be high even in case of a pure-by-chance participation, because people are more inclined to participate if they live in places where a certain phenomenon potentially occurs, while they are less interested if they are not affected. In conclusion, although more than 98% of the respondents already know what SLR is, as confirmed by the answer to a specific question on the survey form (see Supplementary S1), we believe that such a percentage is not biased due to the way respondents have been selected. In fact, it must be remarked that even the answers provided by people that declared to be familiar with the phenomenon were wrong, although 58% of the respondents live close to the sea. This issue will be discussed in the last part of the paper. Figure S2 in the Supplementary Material shows pie charts describing the age, education and job position of the respondents. Table 2 summarizes these data. We did not ask for gender to avoid any discrimination; however, we believe that for our study the

information would be redundant since the attitude to mitigation and proactive actions does not depend on sex.

**Table 1.** Number of respondents for each country. The four countries involved in the project are shown in red. The total number of completions is 1417.

| Country | Number of Respondents |
| --- | --- |
| Italy | 992 |
| Spain | 249 |
| Greece | 56 |
| Cyprus | 38 |
| USA | 20 |
| Germany | 19 |
| France | 11 |
| UK | 5 |
| Norway | 4 |
| Belgium | 3 |
| India | 3 |
| Ireland | 3 |
| Netherlands | 2 |
| Portugal | 2 |
| Algeria | 1 |
| Argentina | 1 |
| Australia | 1 |
| Colombia | 1 |
| Denmark | 1 |
| Jamaica | 1 |
| Israel | 1 |
| Luxemburg | 1 |
| Malta | 1 |
| Panama | 1 |

**Table 2.** Composition of the sample by age, education and employment.

| Age | Education | Employment |
| --- | --- | --- |
| 16–19 9.53% | Middle 8.62% | Teacher 11.58% |
| 20–35 19.90% | High 23.46% | Retired 9.60% |
| 36–51 33.17% | University 36.25% | Student 16.38% |
| 52–64 29.15% | Post Graduate 31.62% | Other 62.43% |
| >64 8.26% | ------------ | ------------ |

## 3. Analysis of the Questionnaires

The first block of the questionnaire aims at knowing how the public obtains information about the SLR. The respondents could input any answer that applied. About 8% of the respondents ticked only one answer; out of these, about 50% claimed that their main source of information is school and/or university. A combined check with degree of education and age confirmed that they are all students. It is encouraging that the topic is treated at school and university, in particular because the analysis of some of the books adopted in the schools of the participating countries pointed out many gaps and mistakes in knowledge about the phenomenon and its consequences. The goal is to understand whether these errors have been transferred to students or have been explained in classroom discussions. In fact, while waiting for editors to update and correct the school texts, there is a need to train teachers with initiatives to improve their knowledge of the scientific aspects of SLR, of its consequences and of the proactive actions to be passed to their students.

The remaining respondents ticked more than one source of information. The analysis of multiple answers about information sources is rather complicated. In fact, in this very case, the total percentage for each information source may be greater than 100%. Thus,

the evaluation must be performed in a qualitative way. Television and internet were the most popular answers, followed by newspapers and magazines. Social networks (which we expressly distinguished from the internet) also have a significant impact. Apparently, our sample did not collect information, or at least very little, from municipalities and local institutions. Figure 2 shows the distribution of answers.

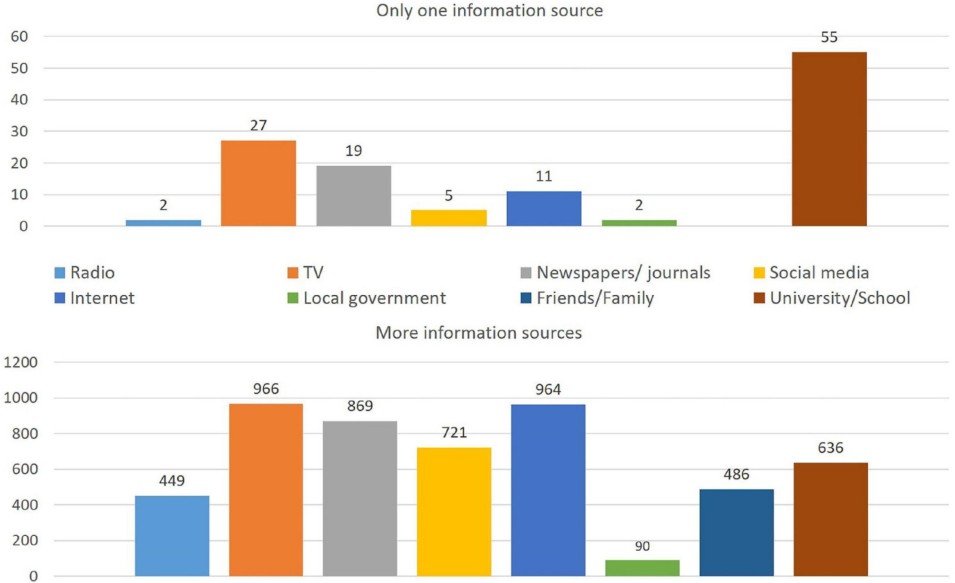

**Figure 2.** How do respondents obtain information about SLR. Upper panel: respondents who input only one answer. Lower panel: more than one choice.

As a general comment, the issue is how reliable and correct the information disseminated by the media is. This is a common problem with other natural hazards or other fields like, for example, medicine. TV shows, internet blogs, articles on newspapers and posts on social networks are often not directly managed by experts. The participation of researchers in TV broadcasts is limited and their presence on social media is often denigrated by haters and keyboard warriors. Sensitive topics are often treated by non-experts. The solution to this issue is to have more people directly listen to experts or to increase the presence of experts in the media. However, academics are not keen, nor do they have experience to present themselves in a "fascinating" way to attract followers on social media. Conferences and round tables, which are the places where scientists come into contact with the public, are considered too complex to understand. Moreover, the presence of experts in the media is dependent on the interest of the public: researchers and experts become popular during or right after a natural disaster, that is, at the worst time to foster prevention, and worse, are never requested during peace time because a particular topic is "not on the news".

The second section is about causes, consequences, responsibilities, actions to mitigate the risk and what each citizen can do to reduce the ongoing SLR. Questions 1, 2 and 5 accept multiple answers, while question 3 and 4 answers use a Likert scale (scores 1 to 5) [27].

For questions 1, 2 and 5 we distinguished respondents who input only one choice (they are, respectively, 12%, 8% and 7%) from those that ticked more options. For question 1, the respondents who expressed only one choice input global warming (66%), ice melting (24%) and subsidence (8%) as causes of the phenomenon. Not only do the respondents seem to have clear ideas by ticking only one answer, but they also indicate what are generally considered the "correct" main causes. It must be remarked that ice melting is a consequence of global warming, so the two answers are different aspects of the same issue. Most of the respondents (1237 out of 1417) input at least two answers. Only very few believe that volcanoes and earthquakes may cause SLR, while the majority declare correctly that the phenomenon originates from global warming, ice melting and subsidence. This latter cause was ticked by fewer people, showing that it is not adequately related to sea level in the

literature and in the media. However, about 70% of the respondents that ticked subsidence as one of the causes (243 out of 345) also chose ice melting and global warming, showing a good knowledge of all the causes of SLR.

For question 2, which was about the consequences of SLR, respondents who marked a single option chose mostly to leave their homes (55%). However, a significant number of participants chose the temperature of the Earth rises (16%), effects of tidal waves are amplified (11%), coastal areas turn into lakes and swamps (11%) and even that thunderstorms become bigger (4%) as being consequences of SLR. Here, the respondents show some confusion between the causes (increase in the temperature of the Earth) and the effects of SLR; nevertheless, they understand that the main threat is to be obliged to abandon their homes to avoid being flooded. In the case of multiple answers, there is again a prevalence of the answer about the abandonment of the place where one lives, followed by issues about harbors and beaches. Surprisingly, the answer about increasing temperature was also chosen by many people in this case.

Finally, for question 5, which had one choice, only two answers were chosen: the majority of respondents gave credit to scientific studies, since about 87% of the respondents stated that the best way to reduce SLR is to adopt science-based solutions; the remaining believe that fostering sustainable mobility is necessary. Those who chose more than one solution distributed their answers over a wide range of chances: they certainly knew that using air conditioners, heaters and private cars is counterproductive, but do not believe that recycling, getting zero km food and saving water could help to fight the issue of SLR.

The core of the survey comprised questions 3 and 4. The first aimed at knowing who is more responsible, or who has more capacity, for mitigating the SLR. The respondents had to attribute a score from 1 (more important) to 5 (less important) to 5 categories involved, at different levels, in the issue of the SLR. The categories were scientists, engineers, government representatives, schools and citizens.

According to our sample, citizens are the least responsible, while governments are the most responsible. The issue here is not that central or regional governments are blamed for major impacts to the environment and for anthropogenic climate change effects. They are responsible for allowing the construction of homes, infrastructure and buildings near the coastline, without securing a buffer zone against coastal floods. However, this is a striking result as it clearly highlights perception gaps and needs. The gap is that citizens believe they have neither the responsibility nor power to mitigate this disruptive trend. The needs concern the empowerment of citizens to fight top-down decisions that increase rather than mitigate such disruptive trend.

The percentage of answers that free citizens from any responsibility is high, reaching 73%. Governments were assigned the largest burden with more than 50% of answers in position 1 (most important), followed by scientists (29%). The trend of these values was also very similar when answers were subdivided by respondents, employment or age. Figures 3 and 4 show histograms relative to the distribution of answer to question 4 for the whole sample (Figure 3) and for sub-groups according to employment (Figure 4). These groups may be considered as the position held in the society. Students, who have no experience yet, tend to equally subdivide the charge in all categories, the only exception being the belief that citizens are powerless against SLR. It is remarkable that teachers rated school as less important than students did. This means that students have more expectations than what teachers believe they can do.

Finally, it is noteworthy that the respondents believe that scientists are, at the same time, responsible for the current situation (29% of answers on the question about responsibilities) and a resource to solve the issue (adopting science-based solutions).

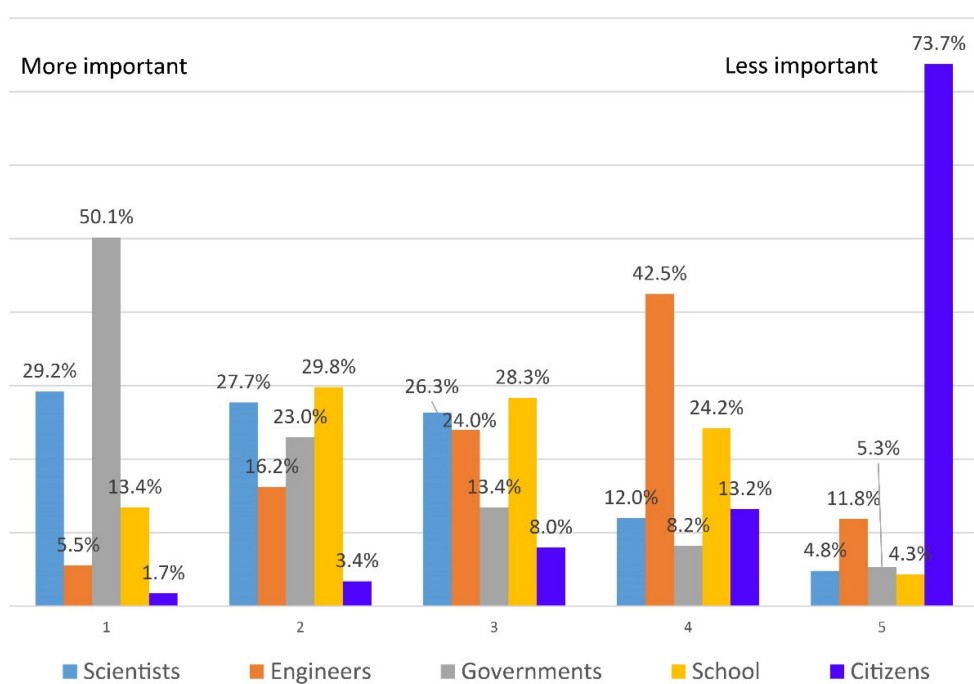

**Figure 3.** Answers to the question "who should primarily work to reduce the damage caused by rising sea level".

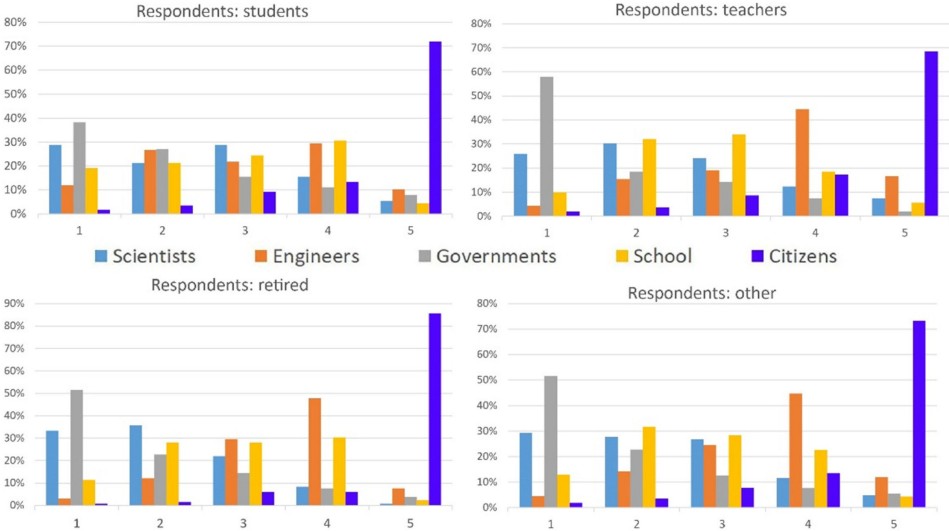

**Figure 4.** Answers to the question "who should primarily work to reduce the damage caused by rising sea level" divided for sub-groups according to employment.

Question 4 asks the respondents to rate, from fundamental to useless, actions to adapt cities to the rising sea level. The questionnaire accepts more answers for the same rating (e.g., more than one proposal could be rated fundamental). Figure 5 shows the answers to the questionnaire. The total number of entries is 1417 times 6, so each histogram bar may have a size greater than the number of respondents.

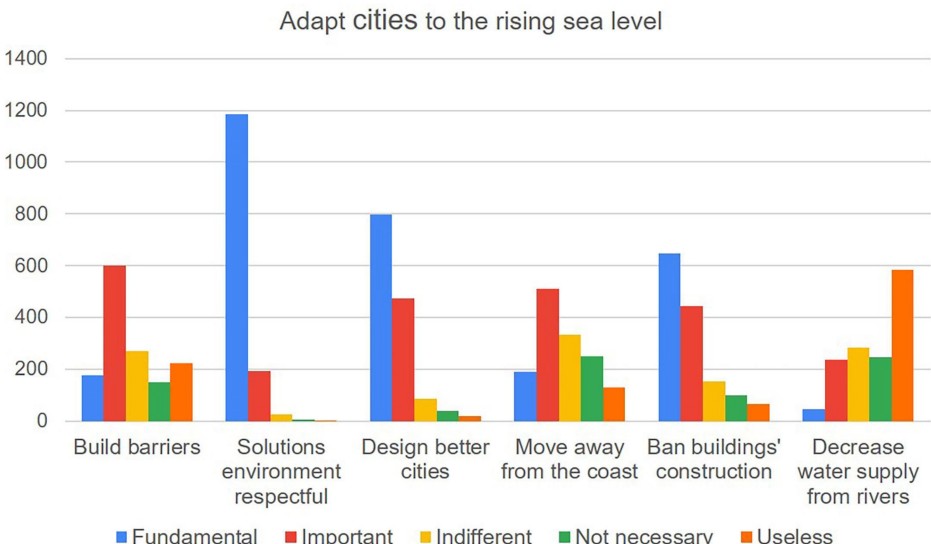

**Figure 5.** Answers to the question "what we need to do for our cities to adapt to the rising sea level effects".

As a general comment, our sample believes that temporary solutions, like building barriers, are not feasible or satisfactory. Some of the respondents consider it not necessary (150 answers) or even useless (222). The majority believes that it is fundamental to have more respect for the environment and to build cities that take into account a "green" view, including avoiding construction on coastal areas. In practice, our sample bet on a better future more than on protection of the existing infrastructures. It is remarked that about only half of the respondents are "ready" to move away from the coast. In fact, 697 respondents declare it is fundamental or important to leave the coast; 332 believe it is indifferent, while 379 think it is not necessary or useless. Out of these, conversely to what was expected, it is not the older respondents who would not leave their place but the "middle aged" ones (33% of those who chose useless or not necessary were aged 36–51). By looking at the overlap between the two solutions (to leave the coast and build up barriers), it was found that about 200 people believe that it is fundamental or important to build barriers and useless or not necessary to leave the coast. In other words, they would be ready to take a reasonable risk by carrying on living in the same place by protecting themselves with defensive barriers. Finally, it is noteworthy that most of those who would not leave the coast are resident there (65%).

## 4. Discussion

Despite the limitations due to the way the questionnaire has been administered and the number of answers, the analysis of the results shows, for the first time ever to our knowledge, a frame of the current perception of the public on SLR and its consequences along the coasts. The random sampling scheme adopted in the collection of the questionnaires is reflected in the diverse amount of responses for each category of participant. Generally speaking, it would be recommended to have similar numbers in each category to make comparisons among the answers and to speculate about the different uncertainties and shortcomings of each age, education or employment category. However, the aim of our survey was to investigate the general public opinion and knowledge in regard to SLR. In fact, the categories themselves were very wide and suited to different sizes of groups of respondents. The goal of interacting with a large audience has thus been achieved, and the results of the survey, although not conclusive, highlight gaps and the need to calibrate future educational activities to foster awareness and possibly proactive actions on the SLR. The main findings from the analysis of the questionnaire are therefore discussed. In a few cases, we also point out different attitudes of the diverse categories, but the reader should bear in mind that these are only qualitative speculations, given what was stated above about the size of each category.

The problem of SLR is already known by some of the people involved in the investigation. This knowledge is shared by both coastal populations and those living in inner regions far from the sea, for whom the issue does not represent a pressing threat. The public is informed through traditional media, in particular through television. This concerns especially the older part of the sample, who also read newspapers and magazines. It is known that access to such traditional media is generally unevenly distributed among the population: older people are generally more familiar with the printed press than young people. Younger people get their information mainly from the internet and social media, where the spread of fake news and inaccurate or confusing information is mostly uncontrolled. Our sample shows that local administrators and schools play a secondary role in the communication to the public. Given that they are a reference for students and citizens both in everyday life, but especially in emergency situations, an effort must be made to render them a reliable source for information and dissemination. The need to make "institutions" become a reference for information was already evident before our survey. It must be remarked that "expert" opinion as a source of information is not limited to traditional media, where, as discussed above, scientists are already present. The challenge is to transfer experts to realities where they may have a larger public but in which they have no, or not enough, experience.

As for the causes, a part of our sample is aware of the phenomena that contribute to SLR, and a part of the respondents already know the phenomenon of land subsidence. This is comforting because it's a complex concept to understand, with slow and hard-to-observe effects. The numbers suggest that insisting on the subject of land subsidence in schools and textbooks is crucial. In fact, although there are at the same time "land lifting" on the globe which mitigate global warming sea level rising in some areas [24], subsidence it is one of the factors that accelerate SLR but it is still unknown to many people.

Regarding the consequences of SLR, the sample shows confusion between causes and effects. Moreover, there are only a few cases in which the compilers have indicated all the possible consequences and this again shows that a careful work of education is necessary to describe the impact, both social and economic, of a phenomenon which is penalized in terms of perception, by a relative low velocity compared to other natural disasters. In this regard, some compilers linked the SLR to the occurrence of other phenomena, for example, earthquakes. Although this is a small percentage of people, it indicates a tendency to confuse natural processes by attributing to a common cause events that are profoundly different from each other.

A weak point emerged from the analysis of the answers in the section related to who should work to reduce damage caused by SLR. This is the tendency of a lack of understanding of one's own role in reducing the phenomenon. The whole sample, with small differences between the categories (age, occupation, degree), has expectations from local rulers and administrators on adaptation and mitigation actions. In addition, part of the sample believes that it is up to scientists to set up prevention proposals and law enforcement activities. Conversely, the responsibility attributed by the sample to citizens is almost nil. In this, the citizens seem to discard their responsibility and forget that not only individual actions by a large number of people can make the difference but also that politicians and rulers are, or at least should be, sensitive to citizens' requests. These aspects highlight the need to work more and better with citizens on their awareness, an action in which schools can carry out actions aimed at training future citizens and administrators to become more aware and active.

A very interesting point is related to actions to adapt cities to SLR. Compilers tend to be reluctant to move inland from coastal areas, but they are also convinced that temporary solutions, such as the construction of artificial barriers, are of little use. They are convinced that the only way to mitigate SLR is to adopt environmentally friendly solutions, and are largely in favor of banning construction along the coast.

Finally, they are aware of the best practices to adopt daily to contain global warming (cut greenhouse gasses emission, use of sustainable mobility, solutions based on scientific

studies, encourage recycling). To this point it is worth noting there is a growing environmental awareness in the population, particularly in the new generations, also thanks to international initiatives such as the Conferences of the Parties and the Paris Agreement that aim to a climatic neutrality by 2050 (https://climate.ec.europa.eu/index_en accessed on 6 September 2023). Although there are denialist positions on global warming and SLR, scientific data nevertheless agrees in showing a continuous and growing trend of rising temperatures and sea levels at a global scale.

**5. Conclusions**

The analysis of 1417 responses to the questionnaire from 23 countries showed that the investigated sample has good basic knowledge of SLR. In some cases, however, citizens who directly experience SLR (like those living in exposed areas) have gaps and preconceptions that must be eradicated. In addition, it is necessary to better inform and educate citizens, points on which the whole sample reached a very small number of "correct" answers. These concern the scientific aspects of the phenomenon, the role of land subsidence in exacerbating the effects of SLR and the behavioral aspects of the need to foster awareness that each citizen can play against global warming and, subsequently, SLR. In both cases, a greater collaboration between scientists and schools must be strengthened, with projects and educational programs that help students and teachers to see climate change in all its nuances, of which SLR is one of the related aspects, reminding citizens that these are interconnected phenomena. It is also necessary to collaborate with publishing houses, because a recent analysis has highlighted strong deficiencies in the description of the causes and effects of SLR on school textbooks for middle school level. For example, the topic of subsidence is rarely treated, and if it is, it is not adequately described. Moreover, it is time to include in school books description of the causes and consequences of SLR as a separate geologic–climate topic and not as a simple secondary effect of climate change. Finally, the sample we analyzed concerns only a part of the population that for is interested in natural phenomena. In an attempt to involve more people in an educational program, other actions should be considered, including adapting the technical language to an even less experienced audience and extending the collection of data to social media networks.

**Supplementary Materials:** The following supporting information can be downloaded at: https://www.mdpi.com/article/10.3390/geohazards4040021/s1, Figure S1: Questionnaire on sea level rise; Figure S2: Pie charts of the job, age and education of the respondents.

**Author Contributions:** Conceptualization, S.S., E.E., G.M., M.D.L. and M.A.; methodology, S.S.; formal analysis, S.S. and E.E.; data curation, S.S., E.E., G.M., M.D.L. and M.A.; writing—original draft preparation, S.S., E.E., G.M., M.D.L. and M.A.; supervision, S.S. All authors have read and agreed to the published version of the manuscript.

**Funding:** This study has been funded by the European Union through the Directorate-General for European Civil Protection and Humanitarian Aid Operations (DG ECHO) by the SAVEMEDCOASTS-2, Grant Agreement No. 874398. "www.savemedcoasts2.eu (accessed on 20 September 2023)".

**Institutional Review Board Statement:** Not applicable.

**Informed Consent Statement:** Not applicable.

**Data Availability Statement:** The data presented in this study are available on request from the corresponding author.

**Acknowledgments:** The authors would like to thank Xenia Loizidou, Demetra Orthodoxou and Michael Loizides (ISOTECH, Cyprus), Josè Navarro and Michele Crosetto (Centre Tecnològic de Tele-comunicacions de Catalunya, Spain), Lucia Trivigno and Antonio Falciano (Centro di Geomorfologia Integrata per l'Area del Mediterraneo, Italy), Michele Greco (Fondazione Ambiente Ricerca Basilicata, Italy), Claudia Ferrari and Chiara Tenderini (City of Venice, Italy), Charalampos Georgiadis (Aristotle University of Thessaloniki, Greece), Giovanna Forlenza and Simone Vecchi (Istituto Nazionale di Geofisica e Vulcanologia, Italy) for their valuable contribution for the translation and dissemination of the questionnaires through citizens and online.

**Conflicts of Interest:** The authors declare no conflict of interest.

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
