# Peer review of "Is Sea Level Rise a Known Threat? A Discussion Based on an Online Survey"

_2624-795X, doi:10.3390/geohazards4040021_

Round 1
Reviewer 1 Report
Attached please find my comments

Moderate editing of the English language is required. please follow my editorial-specific comments.
Author Response
We thank reviewer #1 for his/her valuable comments that improved the revised version of the ms. Changes in the manuscript are highlighted in yellow. Below we reply point by point to the comments hoping the revised version is now suitable for publication.
Point by point responses (in italics)
The paper deals with an interesting theme, i.e., society’s knowledge and awareness about the effects of SLR on the coastal zone, but the used methodology (a simple and popularized) questionnaire needs an enhancement of statistical post-processing and more clarifications or discussion of the followed approach, thus, the limitations of the used approach should have been more thoroughly discussed.
We are quite aware of the global impact and cascading effects of sea level rise, rightly stated by reviewer 1. In fact, the SAVEMEDCOASTS and SAVEMEDCOASTS-2 projects, from which this work was born, deal with the phenomenon in its complexity with the related impacts. However, we decided not to include these elements in a questionnaire that we knew could only be carried out online. We felt that introducing more complexity could have distracted from completing the questionnaire itself. For this reason we have preferred to limit our investigation only to the basic aspects of the phenomenon and its causes, on which to stimulate the reader to adopt more aware behaviours.
It is not clear what the innovative input to current research on SLR is, adding permanent value to the relevant scientific literature. The paper feels more like an overall technical repost of the referred project rather than a new research article with novel scientific methodologies or findings. Therefore, regretfully I cannot propose publication of the submitted paper. In the following, I present my review comments/questions.
The goal of our ms is not to provide new insights on the scientific issues related to climate change and sea level rise. Instead it aims to provide new and original data on the knowledge and awareness of the people on SLR and its causes and effects.
1) L.2-3: The funding Project name should not be included in a scientific paper's title. Only research investigation themes and facts are supposed to be mentioned in the Abstract and the paper’s title. Please refer to the funding source and the project/program only in the Acknowledgements.
We have removed the name of the project from the title. However we think that it is not so critical to leave the project’s name in the Introduction to explain to the readers that the survey has been carried out in the frame of a scientific activity integrated in a multidisciplinary project. The name of the project and the details about it have been removed from the rest of the paper.
2) L.9: The authors infer that “since 1880, global warming has been triggering sea level rise at anunprecedented rate”, but based on the following figures (Fig.1; by IPCC, 2013) the SLR ratewas quite low from 1900 to the 30’s, then doubled (yet still quite low; 1-1.4mm/yr) since the90’s, and there is an abrupt increase in the trend (based on remote sensing, not tide gaugedata!) giving a 3.3mm/yr SLR rate. Thus, the “unprecedented” rate could refer only to the last30 years. Moreover, a mean global SLR of totally 15cm within 115 years (1900-2015) may not be considered so alarming. The problem would rather be the projections for the 21st century,where the (still very unlikely) most pessimistic scenarios assess a global SLR of about 1m (from2000 to 2100), most probably not exceeding the 40cm or maybe up to half a meter. Pleasenote that such values of mean SLR by themselves are not considered an immense threat, asthis process is a very slow one and humanity/society will more or less adapt to the coastlineretreat until 2100. The problem is that even these small changes in mean SLR could triggergreat changes to the coastal wave climate and storm surge levels, especially in the case ofextreme weather events that can induce episodic coastal floods with catastrophic impacts onthe coastal environment. Of course, there are coastal sites (such as Venice etc.) that suffer from land subsidence, andthese are bound to be affected by higher local values of SLR, and there, the seawaterencroachment is a huge issue and the permanent inundation effects might have even moredevastating effects to the coastal built and natural settings. Therefore, I would rather tone down the aforementioned overstatements throughout the text.
We have revised the ms including new lines concerning the above issues. We have added one figure (IPCC data for mean SLR past records and future projections)
3) L.12-13: I would prefer calling it coastal inundation rather than flooding, which relates more to fast episodic events induced by high coastal waves or storm surge. See reference of Flick et al. (2012) for a commentary on proper terminology. Flick, R. E., Chadwick, D. B., Briscoe, J., & Harper, K. C. (2012). “Flooding” versus “inundation”. Eos, Transactions American Geophysical Union, 93(38), 365-366. https://agupubs.onlinelibrary.wiley.com/doi/full/10.1029/2012EO380009
We have changed from flooding to coastal inundation. The reference has been added.
4) L.15-16: No need to refer to the funding project in the main text, and much more in the Abstract. This should be stated only in the Acknowledgements. Please keep to the scientific facts only, or else the paper feels like a project’s technical report.
We have removed the name of the project from the title. However we think that it is not so critical to leave the project’s name in the Introduction to explain to the readers that the survey has been carried out in the frame of a scientific activity integrated in a multidisciplinary project. The name of the project and the details about it have been removed from the rest of the paper.
5) L.27-28: This is a rather simplistic representation of a well-known phenomenon that has a lot of determining components (dynamic, steric, atmospheric, ocean water mass-exchange, astronomical/tidal etc.). Please refer to proper literature, e.g., Calafat et al. (2022) or others for a complete review of the issue.
Calafat, F. M., Frederikse, T., & Horsburgh, K. (2022). The sources of sea-level changes in the Mediterranean Sea since 1960. Journal of Geophysical Research: Oceans, 127, e2022JC019061. https://doi.org/10.1029/2022JC019061
We have revised the text and cited the suggested literature. We remind that the paper does not deal with the phenomenon itself but with the perception of the public.
6) L.31-32: what might undergo retreat and erosion is a coastline or shoreline, not infrastructures. Maybe better use the term coastal sites that pertain all the above or something like that.
We have revised the text according to the suggestions.
7) L.33-34: SLR, following the definition of the authors, is meant to be the rise of Mean Sea Level due to climate change effects. Not the episodic sea level increase due to storm surge or wave action on the coast that can cause abrupt and extended flooding that lasts from an hour up to a few days. So, to my mind, SLR is a very slow process of geological timescales, and the changes in coastal retreat can be barely noticeable in typical human experience timeframe, as it takes many decades up to hundreds of years to take effect. SLR can cause extended coastal inundation and seawater encroachment of inland coastal areas but in a very slowly progressing rate, even though the evolution trend seems to have been accelerated the last decades from ~1mm/yr to a few mm/yr. Thus, I wouldn't call it "much more disruptive than earthquakes or volcanic eruptions", that are clearly very abrupt and violent phenomena. Society, in its majority, can and will conform to and cope with the changes in coastline retreat. Thus, the amplification of other processes, such as episodic flooding and erosion due to storm surge and wave extremes, probably enhanced by SLR, are the main issues for coastal management (vulnerability, risk etc.).
Please see our reply to point n.9
8) L43-44: Please note that a SLR of about 1m is the most (highly unlikely) pessimistic (worst case) scenario with the most extreme SLR projection (see Fig. 1 above). The uncertainty (in terms of probabilistic theory datasets) of it should be thoroughly discussed. The most probable projections show a SLR of <50cm for the 21st century, while already being in 2023, the projections have been restricted down to 28-30cm above current MSL.
We have clarified the expected SLR values according to the literature.
9) L.51-52: How is SLR (a very slow process) more disruptive than earthquakes or volcanic eruptions? This is an overstatement (especially after the latest extremely devastating conditions in the Mediterranean area, i.e., the February 2023 Turkey-Syria earthquake).
We are aware that a single earthquake or volcanic eruption may be very destructive. Anyway they affect only limited areas of the Earth's surface even during the strongest events, such as the Turkey earthquake of February 2023. Conversely, SLR is a global phenomenon that can affect in time the coasts of each continent and island of the world, as well as the populations living close to the coastline since historical times (see Benjamin et al., Late Quaternary sea-level changes and early human societies in the central and eastern Mediterranean Basin: An interdisciplinary review. Quaternary International Volume 449, 25 August 2017, Pages 29-57). Because SLR is the cause of coastal erosion, coastal flooding, loss of land and related cascading effects (damages to drainage systems, salification of water tables, agriculture, coastal infrastructures, etc), heavy damages can be expected along the global coasts in the next decades. In some areas SLR is yet causing heavy damages and huge costs for the society, such as in Venice (Italy) or Jakarta (Indonesia). However, we agree that the sentence in the original form was misleading and we changed it.
10) L.58-59: Same here (please see Comment #1).
See responses 1 and 4
11) L.65-73: This belongs to a technical report of a project, not a scientific paper. Please stick to the investigative facts and research setup and change the narrative away from a specific project focus.
We have changed the text according to the suggestions.
12) L.98-100 and References list: The literature review is kind of weak; it should be enhanced with more available references (by a “SLR + Questionnaire” search numerous papers appear, too many to list); please find a few in the following:
- https://www.mdpi.com/2073-4441/9/12/941
- https://link.springer.com/article/10.1007/s10113-012-0399-x
- https://www.tandfonline.com/doi/full/10.1080/13669877.2010.503935?casa_token=H50aLrkAZq4AAAAA%3As0SckIpvFWHSRGvPsLm6K6SziVUTYRqfz7mmYqyLT8pVPADnU78n_PbwIftIxe1cFZ6fhp9J99U
- https://link.springer.com/article/10.1023/A:1009684210570
- https://agupubs.onlinelibrary.wiley.com/doi/abs/10.1002/2015EF000346
- https://www.sciencedirect.com/science/article/pii/S1364815213000558?casa_token=1jmLjTbRxUAAAAA:jQTw0ZmTB1rYkZlgrWPjEuuphQswixdc9L4uA5cfPkgKD40_yxSzq6cRllw7p4f40dZubVYCg
- https://www.tandfonline.com/doi/full/10.1080/14498596.2014.943311?casa_token=fr HOA40E8iYAAAAA%3AZt4SKIsO4NDcwv_KHhfl1oTt1lJN2VlYVy1APEz5Qpk0f3hEGlfFjE7w AlBs1gEo-PrJx16iPqKA
- https://www.mdpi.com/2072-4292/3/9/2029
- https://www.nature.com/articles/nclimate2469
- https://www.sciencedirect.com/science/article/pii/S0964569113001439?casa_token=SF CjZXlvWtoAAAAA:Jp2gOHAX0Nb8nYzer-fZo_LY1xB24Jbm0awlGeVnXP9VWS3CA67biu72-btLBiijLFDZmTxBIA
and many more.
We have cited in the texts part of the suggested references. We thank the reviewer for providing them.
13) L.144-149: it is not clear to the reader how this was performed. Please elaborate on the method to trace outliers and copycats.
We changed the sentences in order to better specify what we did in our survey.
14) L.151-153: Which sampling methodology was used here? Snowball sampling technique or other? Was it convenience sampling or maybe stratified? Sounds like totally random sampling. This should be discussed, and its drawbacks pinpointed in the text.
As somewhat stated in the paper, the sampling was totally random. In fact we needed unbiased sampling, which would not be the case in snowball or convenience sampling. We added a few sentences to explain our choice and what this implies.
15) L.265-266: I do not agree with the authors’ surprise. Climate change is a global scale phenomenon, that requires international large scale actions with a global consensus. Governments, armies, and big companies (industry, shipping business, transportation etc.) are mainly to blame for the major impacts to the environment and for anthropogenic climate change effects. Moreover, central, or regional governments are also responsible for allowing construction of homes, infrastructure, and buildings near the coastline, without securing a buffer zone against coastal floods.
We totally agree with the reviewer’s opinion. The issue here is not that governments are considered as the main responsible for the SLR, which is dealt. The big issue, in our opinion, is that citizens (that live in houses close to the coast, produce a lot of greenhouse gas through the production of waste, use of cars, etc., not caring at all of the environment, thus increasing global warming and SLR) believe that their responsibility in such a situation is nil. We discuss this point in (former) lines 383-387 of the ms : In this the citizens seem to discard their responsibility and forget that not only individual actions by a large number of people can make the difference but also that politicians and rulers are, or at least should be, sensitive to citizens' requests. These aspects highlight the need to work more and better with citizens on their awareness, an action in which the school can carry out actions aimed at training future citizens and administrators to become more aware and active. We added a few lines to better specify why we are “surprised”, hoping to be more clear on what we consider very interesting findings.
16) Figure 3: Provide color scale explanation for bars.
We have included a color scale bar in figure 3.
17) Figure 4: The “decrease water supply from rivers” is a trick question but rephrasing it to something close like “protect river littorals with embankments to reduce fluvial flooding” would be a good deal, in order to reduce the combined effects of compound flooding. It is not very clear, how were the questions were chosen. The authors should elaborate on that.
The suggestion from the reviewer is interesting and we appreciate it. Our question is referred to the effect of river flooding at their mouth in SLR conditions. Infact, the water discharge is prevented in case of high levels of sea water thus becoming a potential risk of flooding for the surroundig river areas that could be protected by embakments along the river course. Unfortunately, the question cannot be rephrased as the data collection is finished. The questions were chosen in order to have a limited selection of topics and phrased in a way that possible misunderstandings by the respondents (i.e. the general public) would be minimized.
18) From the Discussion It is not apparent to the reader what is the new knowledge (with permanent value) added to the relevant technical literature? This has to be clarified.
We have clarified this point in the ms. We do not believe that our analysis is conclusive. However, it is a good starting point for further discussion.
19) L.377-384: The commentary in this part is based on the authors' preconceptions and does not emerge from sound scientific data, but more hinges on speculative analysis that migrates the duty to act against anthropogenic climate change from big players in the “environmental impacts” realm to individuals.
We have clarified this point in the ms which is not based on the author’s perceptions but from scientific data
20) The methodology of post-processing the answers of the questionnaire is kind of weak. There are numerous decision-making software solutions that could assist, PCA method, etc.
All data have been manually processed using algorithm designed for the scope. We believe that, given the amount of data, this procedure better ensures checking their reliability.
Specific Editorial Comments:
1) L.12: correct to worst case climate scenario
2) L.12: please rephrase; term level used twice in the same sentence
3) L.29: correct to low elevation coastal…
4) L.29-30: better correct to islands and littoral urban areas (or cities)
5) L.31: correct to cultural heritage sites
6) L.34: define SLR in brackets next to its first full name mentioning in the main text (maybe L.27?)
7) L.88-91: Please rephrase, the sentence is hard to follow.
8) L.89: replace (NOAA, 2016) with proper number and renumber all refs if necessary.
9) L.92: rephrase, feels like a repetition; action/actionable term written 2 times.
We have revised all points
Reviewer 2 Report
Dear Authors,
The paper is interesting and useful. Abstract, content and conclusion are well written. Figures are correct. Thus I recommend publication of the paper with minor correction.
Minor comments:
Please be careful regarding abreviations. For example SLR and GSLR at line 34 and 44, respectively, are not defined?
Ragarding e.g. citation, in lines 359- 362: „However, the number suggest that it is necessary to insist that land subsidance is treaated in schools and in school books. In fact, although it is one of the main factors that acceleratee sea level rise, it is still unknown to many people“ it should be careful bacause of there are at the same time „land lifting“ on the Globe which mitigate „global warming sea level rising“ in some areas. By land modeling and usinge satellite altimetry (absolute) and tide gauge (relative) sea level observation, GPS measurements and other methods, a dinstinction between „global climate warming“ (e,g. thermal expansion of the ocean water) „non-climate signals“ (e.g. land movements) tried to be made.
References recommended:
Wõppelmann, G.; Marcos, M. Coastal sea level rise in southern Europe and the non-climate contribution of vertical land motion. Journal of Geophysical Research, 11, 2012, C01007, https://doi:10.1029/2011JC007469
Author Response
We thank reviewer #2 for his/her valuable comments that improved the revised version of the ms. All changes are highlighted in yellow. Below we reply point by point (in italics) to the comments hoping the revised version is now suitable for publication.
Dear Authors,
The paper is interesting and useful. Abstract, content and conclusion are well written. Figures are correct. Thus I recommend publication of the paper with minor correction.
Minor comments:
Please be careful regarding abreviations. For example SLR and GSLR at line 34 and 44, respectively, are not defined?
We have revised all abbreviations and acronyms.
Ragarding e.g. citation, in lines 359- 362: „However, the number suggests that it is necessary to insist that land subsidence is treated in schools and in school books. In fact, although it is one of the main factors that accelerate sea level rise, it is still unknown to many people“ it should be careful because of there are at the same time „land lifting“ on the Globe which mitigate „global warming sea level rising“ in some areas. By land modeling and using satellite altimetry (absolute) and tide gauge (relative) sea level observation, GPS measurements and other methods, a distinction between „global climate warming“ (e,g. thermal expansion of the ocean water) „non-climate signals“ (e.g. land movements) tried to be made.
We have taken into account the suggestion
References recommended:
Wõppelmann, G.; Marcos, M. Coastal sea level rise in southern Europe and the non-climate contribution of vertical land motion. Journal of Geophysical Research, 11, 2012, C01007, https://doi:10.1029/2011JC007469
We have cited the suggested reference.
Reviewer 3 Report
This is an interesting paper. But it is more in sociological sciences and I am not sure that GeoHarards include this science direction. however, let the editor decide.
General comments:
1. Introduction - add the citations of real level rise with direct estimatimations (mm/year), which measured by gauges and altimetry and the difference in space from 1880 to 2022 - in present climate.
Please add several citations to threats from wind waves during storms at an increased average level. This is often considered in the joint study of storm surges and wind waves. In the case of global level rise the problem of wind wave is actual.
2. Please add to supplementary files full text of survey questionnaire.
3. Please add the references to sociological methods, which indicate what sample size from which age category and education is sufficient for adequate assessments? 1400 is a lot, but how many of them are aged 30-50 and have higher education? It's enough for adequate assessments?
4. in the discussion section, it is necessary to comments the folowing questions: will humanity be able to stop the temperature growth, stop the melting of ice and the rise in level? Not everyone believes that this process is completely anthropogenic, maybe a cooling will begin? also, not all countries are ready to implement the "Paris Climate Agreement" in the foreseeable future. It is necessary to discuss the hope that the level will not rise.
Minor comments:
1. Abstract: "The worst climate scenario,sea level could rise up to 1.1 m" - in which year? 2100?
2. Keywords: please add : coastal fluds, Mediterranean Coasts
3. In fig2 it is enough numbers in format X.X%
4. In fig3 please add the vertical axis ticks and lables
Author Response
We thank reviewer #3 for his/her valuable comments that improved the revised version of the ms. All changes are highlighted in yellow in the revised version. Below we reply point by point (in italics) to the comments hoping the revised version is now suitable for publication.
This is an interesting paper. But it is more in sociological sciences and I am not sure that GeoHarards include this science direction. however, let the editor decide.
General comments:
- Introduction - add the citations of real level rise with direct estimatimations (mm/year), which measured by gauges and altimetry and the difference in space from 1880 to 2022 - in present climate.
Please add several citations to threats from wind waves during storms at an increased average level. This is often considered in the joint study of storm surges and wind waves. In the case of global level rise the problem of wind wave is actual.
We have updated text and references.
- Please add to supplementary files full text of survey questionnaire.
The questionnaire in english in already published as Supplmentary material S1
- Please add the references to sociological methods, which indicate what sample size from which age category and education is sufficient for adequate assessments? 1400 is a lot, but how many of them are aged 30-50 and have higher education? It's enough for adequate assessments?
We have added a few lines (131-146) to better explain the sampling scheme adopted in the article. Basically, it does not target a given population but the generic public without any sub-selection. To our knowledge, ours is the first survey of the perception of SLR in the mediterranean. Of course, our results will only have a qualitative meaning.
- in the discussion section, it is necessary to comments the folowing questions: will humanity be able to stop the temperature growth, stop the melting of ice and the rise in level? Not everyone believes that this process is completely anthropogenic, maybe a cooling will begin? also, not all countries are ready to implement the "Paris Climate Agreement" in the foreseeable future. It is necessary to discuss the hope that the level will not rise.
We have included some comments on the suggested points in the reviewed version of the ms.
Minor comments:
- Abstract: "The worst climate scenario,sea level could rise up to 1.1 m" - in which year? 2100?
In the Abstract is reported “by the end of this century” that is 2100.
- Keywords: please add : coastal fluds, Mediterranean Coasts
- In fig2 it is enough numbers in format X.X%
- In fig3 please add the vertical axis ticks and lables
We have clarified the above points in the reviewed version of the ms.
Round 2
Reviewer 1 Report
The authors have addressed almost all of my review concerns.
A few text-editing issues remain:
- L.12: correct to "worst case climate scenario"
- In the provided corrected (yellow highlighted) material please watch out the use of period punctuation marks (some double and some missing, especially after brackets of numbered references.
Author Response
We thank reviewer #1 for his/her valuable comments that improved the revised version of the ms. Below we reply point by point to the comments.
L.12: correct to "worst case climate scenario"
We have corrected L12
In the provided corrected (yellow highlighted) material please watch out the use of period punctuation marks (some double and some missing, especially after brackets of numbered references.
We have added period punctuation marks where needed
We have made an editing of English language.